# Curcumin Derivative GT863 Inhibits Amyloid-Beta Production via Inhibition of Protein N-Glycosylation

**DOI:** 10.3390/cells9020349

**Published:** 2020-02-03

**Authors:** Yasuomi Urano, Mina Takahachi, Ryo Higashiura, Hitomi Fujiwara, Satoru Funamoto, So Imai, Eugene Futai, Michiaki Okuda, Hachiro Sugimoto, Noriko Noguchi

**Affiliations:** 1Department of Medical Life Systems, Faculty of Life and Medical Sciences, Doshisha University, 1–3 Miyakodani, Tatara, Kyotanabe, Kyoto 610 0394, Japan; ctuc2020@mail4.doshisha.ac.jp (M.T.); dmp2003@mail4.doshisha.ac.jp (R.H.); sfunamot@mail.doshisha.ac.jp (S.F.); hsugimot@mail.doshisha.ac.jp (H.S.); nnoguchi@mail.doshisha.ac.jp (N.N.); 2Department of Molecular and Cell Biology, Graduate School of Agricultural Science, Tohoku University, 468-1 Aramakiazaaoba, Aobaku, Sendai, Miyagi 980-0845, Japan; kuku.sou99@gmail.com (S.I.); eugene.futai.e1@tohoku.ac.jp (E.F.); 3Green Tech Co., Ltd., 1–7–7 Yaesu, Chuo-ku, Tokyo 103-0028, Japan; m.okuda.pharma8@gmail.com

**Keywords:** Alzheimer’s disease, amyloid-β peptides, curcumin derivatives, glycosylation

## Abstract

Amyloid-β (Aβ) peptides play a crucial role in the pathogenesis of Alzheimer’s disease (AD). Aβ production, aggregation, and clearance are thought to be important therapeutic targets for AD. Curcumin has been known to have an anti-amyloidogenic effect on AD. In the present study, we performed screening analysis using a curcumin derivative library with the aim of finding derivatives effective in suppressing Aβ production with improved bioavailability of curcumin using CHO cells that stably express human amyloid-β precursor protein and using human neuroblastoma SH-SY5Y cells. We found that the curcumin derivative GT863/PE859, which has been shown to have an inhibitory effect on Aβ and tau aggregation in vivo, was more effective than curcumin itself in reducing Aβ secretion. We further found that GT863 inhibited neither β- nor γ-secretase activity, but did suppress γ-secretase-mediated cleavage in a substrate-dependent manner. We further found that GT863 suppressed *N*-linked glycosylation, including that of the γ-secretase subunit nicastrin. We also found that mannosidase inhibitors that block the mannose trimming step of *N*-glycosylation suppressed Aβ production in a similar fashion, as was observed as a result of treatment with GT863. Collectively, these results suggest that GT863 downregulates *N*-glycosylation, resulting in suppression of Aβ production without affecting secretase activity.

## 1. Introduction

Alzheimer’s disease (AD) is characterized by the accumulation of amyloid-β (Aβ) peptides in senile plaques and by intracellular accumulation of hyperphosphorylated tau [1]. An increasing body of evidence indicates that Aβ oligomers are neurotoxic, and trigger synapse dysfunction, memory loss, and cognitive deficits [2]. In the amyloidogenic pathway, Aβ is generated from transmembrane amyloid-β precursor protein (APP) by sequential proteolytic cleavage steps involving β-secretase and the γ-secretase complex [2]. On the other hand, β-secretase consists of a single protein—that is, β-site APP cleaving enzyme 1 (BACE1); γ-secretase is a multimeric complex consisting of four integral membrane proteins—these being presenilin 1 (PS1) or PS2, nicastrin, anterior pharynx-defective 1 (APH-1), and PS enhancer 2 (PEN-2). In an alternative pathway, APP is cleaved by α- and γ-secretases to release the nonamyloidogenic p3 peptide.

Major challenges in AD drug development include achievement of reduction of Aβ production through inhibition of β-secretase or γ-secretase, or achievement of increased Aβ clearance. However, many clinical trials have failed to satisfactorily achieve such challenges due either to lack of efficacy or adverse side effects [3,4]. It has also been suggested that clinical trials should start before appearance of AD symptoms. With regard to the adverse side effects that have been observed, it has been suggested that these may occur as a result of inhibition of secretase, as β- and γ-secretases cleave not only APP but also a number of other substrates [3,4]. For example, γ-secretase has been reported to cleave Notch, cadherins, and CD44. It would therefore be desirable to establish methods for inhibiting Aβ production that do not affect secretase activity.

Curcumin is a yellow-orange natural polyphenol compound which is found in abundance in the rhizome of the plant *Curcuma longa* (turmeric). Several lines of evidence suggest that curcumin has therapeutic effects on AD [5,6,7,8,9]. For example, curcumin can inhibit Aβ oligomer formation and aggregation, and can also inhibit Aβ production [5,6,7]. Several studies have further shown that curcumin can reduce Aβ deposition in the brain and significantly improve cognitive functions in experimental AD models [8,9]. On the other hand, there are a number of clinical trials that report no significant differences in cognitive function in placebo versus intervention groups [10,11,12]. It should be noted that the low solubility in water and poor bioavailability of curcumin have limited its use in clinical trials and in therapeutic applications [6,12]. We were therefore interested in studying whether derivatives of curcumin could be found which might be more effective in treatment of AD than curcumin itself.

Although curcumin derivatives have been the focus of studies seeking to develop inhibitors of Aβ and tau aggregation, and have been the focus of studies seeking to develop imaging probes for detection of Aβ and tau fibrils, there has been very little investigation into whether curcumin derivatives might serve as inhibitors of Aβ production [13,14,15,16,17,18,19]. We previously developed a series of curcumin derivatives and evaluated their inhibitory effects on Aβ production. Of these, we found that CU6/CNB-001 was more effective than curcumin itself in reducing Aβ secretion [13]. We further found that CU6/CNB-001 downregulates intracellular APP trafficking, resulting in suppression of Aβ production in a manner that is independent of secretase activity. Valera et al. has also reported that CU6/CNB-001 promotes Aβ clearance and improves memory in animal models of AD [14]. Although these results might suggest that CU6/CNB-001 should have beneficial effect on AD pathology, we observed that CU6/CNB-001 had little inhibitory effect on the production of Aβ42, which is much more neurotoxic than Aβ40 [13]. 

In the present study, we conducted in vitro screening in an attempt to identify curcumin derivatives that might inhibit Aβ production more effectively than either curcumin or CU6/CNB-001. As a result of screening of curcumin derivatives selected from a library on the basis of similarity in chemical structure to CU6/CNB-001, we found that GT863 reduced production of both Aβ40 and Aβ42. Interestingly, GT863 (formerly referred to as PE859) has been reported to inhibit Aβ and tau aggregation, and to ameliorate cognitive dysfunction, in AD mice models [15,16,17]. Although it has thus been shown that GT863 has beneficial effect in terms of suppressing Aβ aggregation, we were aware of no evidence indicating whether GT863 might suppress Aβ production. Upon finding in the present study that GT863 does suppress Aβ production, we further endeavored to examine the mechanism by which this occurs, demonstrating in the present study that GT863 inhibited Aβ production without affecting β- or γ-secretase activity. We further found that GT863 suppressed protein *N*-glycosylation, including that of γ-secretase subunit nicastrin, and observed that Aβ production was inhibited by mannosidase inhibitors of the type that cause abnormal *N*-glycan processing. We believe that these results provide insight into how GT863 and mannosidase inhibitors might be used to beneficial effect in the context of AD pathologies.

## 2. Materials and Methods

### 2.1. Materials

Curcumin was obtained from Sigma-Aldrich (St. Louis, MO, USA). CU6, GT832, GT855, GT857, GT863, GT934, and GT935 were kindly provided by Green Tech Co., Ltd. (Tokyo, Japan). Dulbecco’s modified Eagle’s medium/nutrient mixture Ham’s F-12 (DMEM/F-12) and DMEM were from Thermo Fisher Scientific (Waltham, MA, USA). Fetal bovine serum (FBS) was from Sigma-Aldrich. Cell Counting Kit-8 was from Dojindo (Kumamoto, Japan). The following antibodies were from commercial sources: APP (6E10), Covance (Princeton, NJ, USA); APP (22C11), Millipore (Billerica, MA, USA); APP (A8717), nicastrin and β-actin, Sigma-Aldrich; N-cadherin, BD Biosciences (San Jose, CA, USA); ribophorin-1 (C-15), Santa Cruz (Santa Cruz, CA, USA); Aβ (82E1) and BACE1, IBL (Gunma, Japan); Notch intracellular domain (NICD, D3B8); myc (9B11), Cell Signaling (Danvers, MA, USA); and low-density lipoprotein (LDL) receptor-related protein 1 (LRP1), Abcam (Cambridge, United Kingdom). All other chemicals, of analytical grade, were from Sigma-Aldrich or Wako (Osaka, Japan).

### 2.2. Cell Lines and Cell Culture

Chinese hamster ovary (CHO) cells that stably expressed human wild-type APP751 (7WD10, CHO-APP) [20] were maintained in DMEM supplemented with 10% heat-inactivated FBS, antibiotics (100 U/mL penicillin, 100 μg/mL streptomycin; Thermo Fisher Scientific), and 250 μg/mL G418 (Nacalai Tesque, Kyoto, Japan). For CHO-APP cells that stably expressed mouse Notch1 ΔE (CHO-APP/NotchΔE) [21], 500 μg/mL hygromycin (Wako) was added to the above culture medium. SH-SY5Y human neuroblastoma cells (European Collection of Cell Cultures, Salisbury, United Kingdom) were maintained in DMEM/F-12 with 10% FBS and penicillin/streptomycin. Cells were grown at 37 °C in an atmosphere of 95% air and 5% CO_2_.

### 2.3. Cell Treatment

Curcumin or curcumin derivative was dissolved in dimethyl sulfoxide (DMSO; Wako) to obtain stock solution (10 mM), and this stock solution was then stored at −20 °C. Each of the various compounds tested were added as DMSO solution to medium. For 24 h treatment, cells were treated with 10 μM curcumin, 10 μM curcumin derivative, or 3 μM GT863. For 48 h treatment, cells were treated with 0.5–3 μM GT863 for 24 h, following which an additional 0.5–3 μM GT863 was added without changing the medium until 48 h. To confirm γ-secretase activity, cells were treated with 1 μM N-[N-(3,5-difluorophenacetyl)-L-alanyl]-S-phenylglycine t-butyl ester (DAPT) (Wako) for 24 h. To inhibit *N*-glycosylation, cells were treated with 1 μg/ml kifunensine (Cayman Chemical Company, Ann Arbor, MI), 1 μg/ml swainsonine (Cayman), or 1 mM *N*-butyl-deoxynojirimycin (NB-DNJ, Wako) for 48 h.

### 2.4. Assessment of Cell Viability

To assess cell viability, 2-(2-methoxy-4-nitrophenyl)-3-(4-nitrophenyl)-5-(2,4-disulfophenyl)-2H-tetrazolium (WST-8) assay was performed using a Cell Counting Kit-8, as described previously [22]. After 22 h of curcumin derivative treatment, medium was replaced with fresh medium because curcumin has a yellowish pigment, following which WST-8 was added, and this was allowed to stand for an additional 2 h before measurements were conducted. Fluorescence intensity was measured at a wavelength of 450 nm.

### 2.5. ELISA for Aβ

Conditioned media was collected and cleared of cell debris by centrifugation. Secreted Aβ was measured by standard sandwich ELISA using human amyloid β (1–40) (FL) and (1–42) assay kits (both from IBL).

### 2.6. Preparation of Cell Samples and Immunoblotting

Whole cell extract and conditioned media were prepared and analyzed by immunoblotting, as described previously [13,23]. Human APP expressed in CHO cells was detected by 6E10 antibody. Human Aβ was detected by 82E1 antibody. C83 and C99 were detected by A8717 antibody. Secreted APP—APPsα or total APPs—was detected by 6E10 or 22C11 antibody, respectively. Endoglycosidase H (Endo H) and peptide-N-glycosidase F (PNGase F) treatments were performed according to the manufacturer’s instructions (New England Biolabs, Beverly, MA, USA).

### 2.7. In Vitro Secretase Assay

In vitro γ-secretase assay using 3-[(3-cholamidopropyl)dimethylammonio]-2-hydroxypropanesulfonate (CHAPSO)-soluble microsomal fraction of CHO cells or yeast was performed, as described previously [21,24,25]. Immunoblotting was used to detect Aβ and NICD. Activity of α- or β-secretase was measured using a commercial kit (ANASPEC, Fremont, CA, USA). To measure total cellular α- or β-secretase activity, an equal amount of cellular protein (40 μg) was used.

### 2.8. Statistical Analysis

Data are reported as mean ± SD of at least three independent experiments unless otherwise indicated. The statistical significance of the difference between determinations was calculated by analysis of variance using ANOVA, Tukey–Kramer test for multiple comparisons and Student’s *t*-test for comparison of two means. The difference was considered significant when the *P*-value was <0.05.

## 3. Results

### 3.1. GT863 was Identified as a Potent Inhibitor of Aβ40 and Aβ42 Production in CHO-APP Cells.

To assess the effects of six kinds of curcumin derivatives (Appendix A) on APP processing, we used CHO cells that stably expressed wild-type human APP751 (CHO-APP) [20]. We first performed a WST-8 assay to evaluate the effect of curcumin derivatives on cell viability, with the results showing that 0.5–20 μM GT832, GT855, or GT857 did not affect cell viability, but that an amount greater than 15 μM of GT934, 5 μM of GT935, or 4 μM of GT863 had a significant cytotoxic effect (Figure 1a). Because we had previously found that 10 μM CU6 for 24 h treatment showed an inhibitory effect on Aβ production in CHO-APP cells [13], we evaluated the effect of curcumin derivatives on Aβ production at the noncytotoxic concentrations of 10 μM for GT832, GT855, GT857, or GT934, and 3 μM for GT863 or GT935. We performed immunoblot assay (data not shown) and ELISA for Aβ40 or Aβ42 (Figure 1b), the results showing that there was a substantial decrease in Aβ40 and Aβ42 secretion in cells treated with GT863 for 24 h, but no significant reduction of Aβ in cells treated with curcumin, CU6, or any of the other derivatives tested. We therefore conducted further experiments to explore the effect of GT863 on APP metabolism.

### 3.2. Treatment with GT863 for 48 h Reduced Aβ40 and Aβ42 Production and Increased C83 and C99 Levels in CHO-APP Cells

We next investigated the effect of long-term treatment with 0.5–3 μM GT863 on Aβ production in CHO-APP cells. The chemical structure of GT863 is shown in Figure 2a. GT863 treatment for 48 h resulted in significant reduction of both Aβ40 and Aβ42 secretion in a dose-dependent manner (Figure 2b) without affecting cell viability (data not shown). The IC_50_ value for Aβ42 secretion was 1.7 μM. Under the conditions tested, levels in whole cell lysate of full-length APP as well as secreted APP, APPsα, and APPsβ—these latter two respectively being the products of α- or β-secretase cleavage of APP—were unchanged as compared with the DMSO control (Figure 2c). On the other hand, levels of APP C-terminal fragments (CTFs) C83 and C99—these respectively resulting from cleavage by α- and β-secretase—were significantly increased by long-term GT863 treatments as compared with the DMSO control. Because both C83 and C99 serve as substrate for cleavage by γ-secretase, these data indicate that treatment with GT863 caused a decrease in Aβ production, not as a result of inhibition of α- or β-cleavage of APP, but as a result of inhibition of γ-cleavage of C83 and C99.

### 3.3. GT863 Did Not Inhibit γ-Secretase Activity in Vitro

To investigate whether the decrease in Aβ that was observed was the result of inhibition of γ-secretase activity by GT863, we examined the effect of GT863 on in vitro γ-secretase activity. We first performed in vitro γ-secretase assay using the CHAPSO-solubilized microsomal fraction of CHO cells and purified recombinant γ-secretase substrates (C99-FLAG and Notch-FLAG), which allows free collisions to occur between enzyme and substrates in solution [24]. Note that NICD is a γ-secretase cleavage product of Notch-FLAG. In a control experiment, co-incubation of the well-established γ-secretase inhibitor DAPT suppressed both the production of Aβ and NICD as compared with vehicle (Figure 3a, lanes 1–3). In contrast to what was observed with treatment with DAPT, we found that co-incubation with GT863 did not show significant reduction either Aβ or NICD production (Figure 3a, lane 4; Figure 3b). 

We further investigated the effect of GT863 on in vitro γ-secretase activity using CHAPSO-solubilized microsomal fractions of yeast transformants expressing the four subunits of γ-secretase (wild-type PS1, nicastrin, APH-1, and PEN-2) and the APP-based substrate C55 [25]. When the yeast microsomes were incubated with phosphatidylcholine (PC), Aβ production was observed (Figure 3c, lanes 2–4), with it further being observed that this Aβ production could be inhibited by co-incubation with γ-secretase inhibitor, DAPT, or L-685,458 (Figure 3c, lanes 5 and 6). Similar to our observations in Figure 3a, co-incubation with GT863 did not affect Aβ production (Figure 3c, lanes 8 and 9; Figure 3d). Taken together, these results suggested that GT863 did not cause inhibition of γ-secretase activity in vitro.

### 3.4. GT863 Inhibited γ-Cleavage in γ-Ssecretase Substrate-Selective Fashion in CHO-APP/NotchΔE and SH-SY5Y Cells

To further investigate the mechanisms by which GT863 affects γ-secretase-mediated cleavage, we examined the effect of GT863 on γ-cleavage in CHO-APP/NotchΔE cells. NotchΔE is a truncated Notch1 protein that lacks the majority of the extracellular domain, and is a direct substrate of γ-secretase. In a control experiment, treatment with DAPT caused decrease in production of Aβ and NICD, and increased in the levels of C83, C99, and NotchΔE, as compared with vehicle (Figure 4a,b). Although treatment with GT863 caused decreased production of Aβ and increased C83 levels, the effects of GT863 were less remarkable than those of DAPT. In contrast to the decrease in production of Aβ that was observed as being caused by GT863, we observed no significant difference in NotchΔE cleavage relative to vehicle control in GT863-treated cells (Figure 4b).

We next evaluated the effects of GT863 on Aβ production and γ-cleavage in SH-SY5Y human neuroblastoma cells that endogenously expressed APP. Similar to the results obtained with CHO-APP cells, we found that GT863 exhibited an inhibitory effect on Aβ40 secretion in SH-SY5Y cells (Figure 4c). The amount of Aβ42 in conditioned media was below our detection threshold (12.5 pg/ml) in SH-SY5Y cells. Using SH-SY5Y cells, we further examined whether there was cleavage of N-cadherin—N-cadherin being cleaved by α-secretase, and the CTF that is generated thereby being subsequently cleaved by γ-secretase [26]—finding as a result that N-cadherin CTF levels increased in cells treated with DAPT but not in cells treated with GT863 (Figure 4d). Collectively, these results suggest that the inhibitory effect of GT863 on γ-cleavage displays specificity with respect to γ-secretase substrate.

We also examined the effects of GT863 on α- and β-secretase activities in SH-SY5Y cells. Using in vitro enzymatic activity assays for α- and β-secretases, we found that there was no difference in α- or β-secretase activity in homogenates of cells treated with or without GT863 (Figure 4e,f). Taken together, these results suggest that the decrease in Aβ production that was observed to occur upon treatment with GT863 was not due to direct inhibition of β- or γ-secretase activity.

### 3.5. GT863 Suppressed N-Glycosylation of Proteins Including Nicastrin

Because GT863 showed different effects depending on γ-secretase substrate, we examined the expression levels of nicastrin, as nicastrin is known to serve as a substrate receptor within the γ-secretase complex [27,28]. Nicastrin is a type I transmembrane protein with a large, heavily glycosylated ectodomain, and has been known to undergo a maturation process in the endoplasmic reticulum (ER) and the Golgi [29]. Human and Chinese hamster (*Cricetulus griseus*) nicastrin have 16 and 12 potential *N*-glycosylation sites, respectively. Although mature nicastrin manifests as a high molecular-mass band on SDS-PAGE and includes complex and high-mannose *N*-linked oligosaccharides, immature nicastrin manifests as a comparatively low molecular-mass band on SDS-PAGE and includes *N*-glycans that have not been subjected to complex glycosylation [29]. As shown in Figure 5a–c, we found that treatment with GT863 caused decrease in mature nicastrin levels (Figure 5b) and increase in immature nicastrin levels (Figure 5c). To determine whether the immature nicastrin band that had been observed to increase in intensity was sensitive to Endo H—with Endo H being specific for *N*-linked chains having high mannose and hybrid-type glycans—whole cell lysates were subjected to digestion with Endo H or PNGase F—with PNGase F being specific for all *N*-linked glycans. The results showed that the lower band that had been observed as increasing in intensity in GT863-treated cells was sensitive to digestion by both Endo H and PNGase F, resulting in the observation of a fully deglycosylated band at about 80 kDa (Figure 5d). Together, these results demonstrated that there was accumulation of immature nicastrin in GT863-treated cells. 

We further evaluated the glycosylation state of nicastrin by comparing the effect of several *N*-glycosylation inhibitors (kifunensine, a potent inhibitor of α-mannosidase I; swainsonine, a potent inhibitor of α-mannosidase II; and NB-DNJ, a potent inhibitor of α-glucosidase), finding as a result that the electrophoretic mobility of the immature nicastrin band that has been observed to increase in intensity in GT863-treated cells was similar to the electrophoretic mobility observed in kifunensine- or swainsonine-treated cells, but dissimilar to the electrophoretic mobility observed in NB-DNJ-treated cells (Figure 5e). We also found that the electrophoretic mobility of low-density lipoprotein (LDL) receptor-related protein 1 (LRP1) and BACE1—which are also *N*-glycosylated proteins—was altered in cells treated with GT863 in a similar fashion, as observed with kifunensine or swainsonine treatment. Together, these results suggested that GT863 inhibited the mannose trimming step but not the glucose trimming step of protein *N*-glycosylation.

### 3.6. Inhibition of Mannosidase Suppressed Aβ Production Without Affecting NotchΔE Cleavage.

We then took a biochemical approach to evaluate the effect of mannosidase inhibition on Aβ and NICD production. Treatment with kifunensine or swainsonine caused reduction in mature nicastrin levels (Figure 6a,b). Under the condition tested, Aβ secretion was significantly reduced, whereas NICD production was unaffected in cells treated with either kifunensine or swainsonine (Figure 6c) as observed in cells treated with GT863 (Figure 4b). These results suggested that the inhibition of *N*-glycosylation, especially the mannose trimming step, suppressed Aβ production selectively. 

## 4. Discussion

Curcumin is a symmetric molecule (Appendix A) in which there are three chemical entities, that is, two phenolic ring systems containing *o*-methoxy groups that are connected by a seven-carbon linker consisting of an α,β-unsaturated β-diketone moiety. Because the diketone of curcumin is substituted by a pyrazole moiety in CU6/CNB-001, we previously speculated that the pyrazole moiety of CU6/CNB-001 may play a role in inhibiting Aβ production [13]. Among the compounds tested in the present study, the pyrazole moiety is present in GT863, GT855, and GT857. Of these, however, only GT863 showed an inhibitory effect on production of both Aβ40 and Aβ42. One of the hydrogens in the pyrazole moiety of GT863 is substituted by a benzene or nitrobenzene group in GT855 or GT857, respectively. We further note that although GT863 and GT832 differ only with respect to whether a pyrazole moiety or a diketone moiety is present, of these two compounds, only GT863 was observed to suppress Aβ production. Collectively, these observations are believed to confirm our speculation that the pyrazole moiety of GT863 may indeed play an important role in inhibiting Aβ production. We also speculate that the good inhibitory effect of GT863 on Aβ production as compared with CU6/CNB-001, GT855, or GT857 may have been due to presence of the hydrogen in the pyrazole moiety of GT863 at the location where there is a benzene or nitrobenzene group in the pyrazole moiety of CU6/CNB-001, GT855, and GT857. Regarding bioavailability, it has been confirmed that GT863 can cross the blood–brain barrier, GT863 being found in the brain for 24 h following oral administration in mice [15]. Furthermore, indications of drug toxicity such as weight loss were not observed in GT863-treated mice during a 6 month administration period [15], indicating the safety and tolerability of oral constant administration of GT863. Moreover, it has been demonstrated that GT863 treatment can reduce aggregated tau and delay onset and progression of motor dysfunction in human P301L tau transgenic mice [15], and can also inhibit both Aβ and tau aggregation, as well as ameliorate cognitive dysfunction, in senescence-accelerated mouse prone 8 (SAMP8) mice [17]. On the basis of our present study, we speculate that inhibition of Aβ production by GT863 may partly explain the beneficial effects on AD pathology that have been observed in in vivo experiments. Further experiments using AD mouse models should be performed to explore this possibility.

Glycosylation is one of the commonest types of post-translational modification observed to occur in proteins. Most commonly, a glycan is covalently attached to either an asparagine (*N*-glycan) or serine/threonine (*O*-glycan) residue of the protein. The functional properties of the *N*-glycan portion of a glycoprotein may include protein folding, trafficking, secretion, enzyme activity, and interaction with other biological molecules [30,31]. *N*-glycosylation begins in the ER with the transfer of a precursor oligosaccharide from dolichyl-pyrophosphate to the luminal side of a polypeptide chain. After the transfer of the oligosaccharide, the *N*-glycan is gradually trimmed by glucosidases and mannosidases in the ER and Golgi during protein folding. Subsequent processing steps occur in the Golgi through the actions of various glycosyltransferases, resulting in a wide variety of complex types of *N*-glycan structures. Because we found in the present study that treatment with GT863 induced accumulation of Endo H and PNGase F-sensitive immature nicastrin, which had similar electrophoretic mobility on SDS-PAGE as was observed in kifunensine- or swainsonine-treated cells, we consider it to be plausible that GT863 operates by inhibiting the mannose trimming step. Furthermore, because we found in the present study that treatment with GT863 inhibited *N*-glycosylation of not only nicastrin but also of LRP1 and BACE1, we consider it possible that GT863 may have inhibitory properties against any of the various enzymes involved in the mannose trimming step (e.g., α-mannosidase I or II, or *N*-acetylglucosaminyltransferase I). Further studies are needed to clarify the effect of GT863 on enzymatic activities in the mannose trimming step.

We furthermore demonstrated that not only GT863 but also kifunensine and swainsonine can inhibit Aβ production without affecting NICD production by γ-secretase. These results suggest that the mannose trimming step within the protein *N*-glycosylation pathway may play an important role in Aβ production. Because GT863 treatment induced an increase in C99 (and C83), the decrease in Aβ that we observed might have been due to suppression of the γ-cleavage step but not of the β-cleavage step. Although it was not clear which glycoprotein is responsible for the GT863-induced or mannosidase inhibitor-induced decrease in Aβ production that we observed, nicastrin is thought to be a likely candidate on the basis of its availability as a γ-secretase substrate, as nicastrin has been shown to function as a γ-secretase substrate receptor [27,28]. Recent papers have further shown that nicastrin serves as a molecular gatekeeper for γ-secretase substrates, sterically blocking longer substrates from gaining access to the active site on PS [32,33]. Because mature nicastrin primarily associates with the active γ-secretase complex [34], and because complex oligosaccharides are not required for the binding of nicastrin to PS [35], it is possible that inhibition of nicastrin maturation might affect substrate recognition without affecting γ-secretase activity. In relation to our present observations, Moniruzzaman et al. reported that Aβ production was reduced in mutant CHO cells that lacked *N*-acetylglucosaminyltransferase I, resulting in increase in immature nicastrin [24]. On the other hand, several other papers have shown that inhibition of α-mannosidase I by kifunensine resulted in increased levels of immature nicastrin but did not affect Aβ production [35,36]. Though there is no clear explanation for this discrepancy, one explanation might be that our present study used CHO cells that stably expressed wild-type human APP (APPwt), whereas the aforementioned papers [35,36] used HEK293 cells that stably expressed Swedish mutant APP (APPswe). Indeed, in support of this possible explanation, we also confirmed that kifunensine treatment did not affect Aβ production despite the fact that it did suppress nicastrin maturation in CHO cells that stably expressed APPswe (Appendix A). Several lines of evidence indicate that although APPwt is processed by β- and γ-secretases in acidic endosomal compartments originating at the cell surface, β-cleavage of APPswe occurs before it reaches the plasma membrane in the secretory pathway [37,38,39]. Furthermore, it has been reported that lipid rafts in post-Golgi and endosome membranes serve as the platform for amyloidogenic processing of APP [40,41,42]. There is therefore a possibility that the glycosylation state of nicastrin may affect substrate recognition by γ-secretase differently in different membrane microdomain environments. Another possibility that cannot be excluded is the possibility that the inhibitory effect of GT863 and mannosidase inhibitors on Aβ production that we observed might have been due to inhibition of or mediation by some other protein(s) and/or some aspect of *N*-glycosylation thereof. In this regard, there is growing evidence that *N*-glycans may play a significant role in AD development [30,31]. It should furthermore be noted that not only nicastrin but also APP, BACE1, α-secretase a disintegrin and metalloproteinase (ADAM10), and neprilysin are all known to be *N*-glycosylated. It is also of note that terminal sialyation of APP *N*-glycans has been reported to enhance Aβ production [43]. Further experiments will be required for in-depth analysis of these points.

In conclusion, we report that the curcumin derivative GT863 inhibits Aβ40 and Aβ42 production without affecting secretase activity. We also report that GT863 suppresses γ-cleavage in a γ-secretase-substrate-dependent manner. We further report that inhibition of the mannose trimming step in the protein *N*-glycosylation pathway is involved in inhibition of Aβ production by GT863. In addition to its ability to inhibit aggregation of both Aβ and tau, we found that GT863 also had a beneficial effect in terms of its ability to suppress Aβ production. Our findings also demonstrate that mannosidase inhibitor can suppress Aβ production without inhibiting NotchΔE cleavage, suggesting the possibility that inhibition of enzymes involved in the mannose trimming step is a potential drug target for sporadic AD treatment.

## Figures and Tables

**Figure 1 cells-09-00349-f001:**
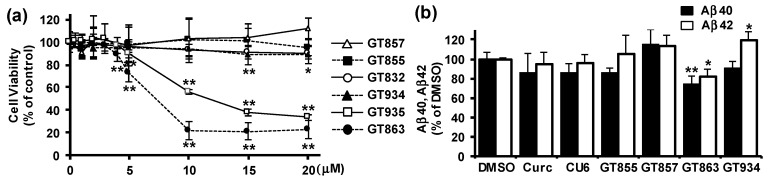
Screening of curcumin derivatives for effect on Aβ production in Chinese hamster ovary–amyloid-β precursor protein (CHO-APP) cells. (**a**) CHO-APP cells were treated with different concentrations of curcumin derivatives for 24 h. Cell viability was measured by 2-(2-methoxy-4-nitrophenyl)-3-(4-nitrophenyl)-5-(2,4-disulfophenyl)-2H-tetrazolium (WST-8) assay. *n* = 3. * *p* < 0.05, ** *p* < 0.01. (**b**) CHO-APP cells were treated with 10 μM curcumin, CU6, GT855, GT857, or GT934, or with 3 μM GT863, for 24 h. Secreted Aβ40 or Aβ42 in conditioned media was measured by ELISA. *n* = 3. * *p* < 0.05, ** *p* < 0.01. DMSO: dimethyl sulfoxide.

**Figure 2 cells-09-00349-f002:**
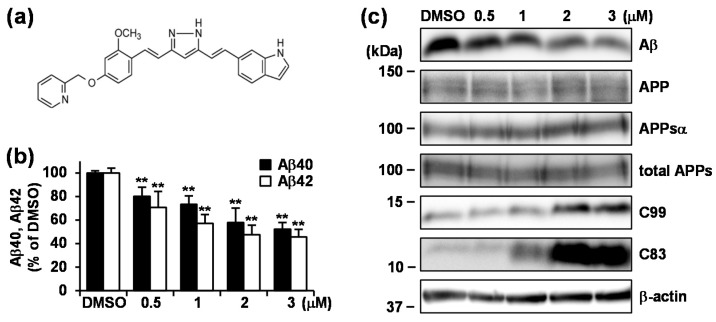
GT863 treatment for 48 h reduced secreted Aβ40 and Aβ42, and increased C99 and C83 levels in CHO-APP cells. (**a**) Chemical structural of GT863. (**b,c**) CHO-APP cells were treated with 0.5–3 μM GT863 for 48 h. (**b**) Secreted Aβ40 or Aβ42 in conditioned media was measured by ELISA. *n* = 3. ** *p* < 0.01. (**c**) Whole cell lysates or conditioned media were immunoblotted with appropriate antibodies as indicated.

**Figure 3 cells-09-00349-f003:**
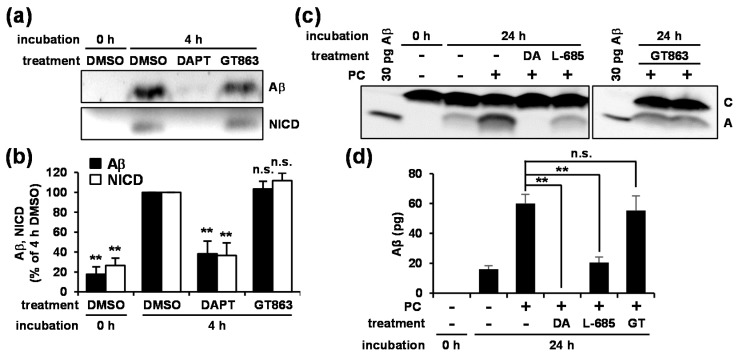
Effect of GT863 on in vitro γ-secretase activity. (**a**) 3-[(3-cholamidopropyl)dimethylammonio]-2-hydroxypropanesulfonate (CHAPSO)-solubilized microsomal fractions of CHO cells and substrates were incubated with or without 1 μM N-[N-(3,5-difluorophenacetyl)-L-alanyl]-S-phenylglycine t-butyl ester (DAPT) or 3 μM GT863 for 4 hat 37 °C. After incubation, reaction mixtures were subjected to immunoblotting with appropriate antibodies specific for Aβ or Notch intracellular domain (NICD). (**b**) Band intensities were quantified by densitometric scanning, relative intensity being shown. Mean ± SD *n* = 3, ** *p* < 0.01, n.s. = not significant. (**c**) CHAPSO-solubilized microsomal fractions of yeast transformants were incubated in the presence of 1 mg/ml phosphatidylcholine (PC) with or without 1 μM DAPT (DA), 1 μM L-685,458 (L-685), or 3 μM GT863 for 24 h at 37 °C. After incubation, reaction mixtures were subjected to immunoblotting. Synthetic Aβ40 (30 pg) was loaded as a marker. (**d**) Band intensities were quantified by densitometric scanning, relative intensity being shown. mean ± SD *n* = 3, ** *p* < 0.01, n.s. = not significant.

**Figure 4 cells-09-00349-f004:**
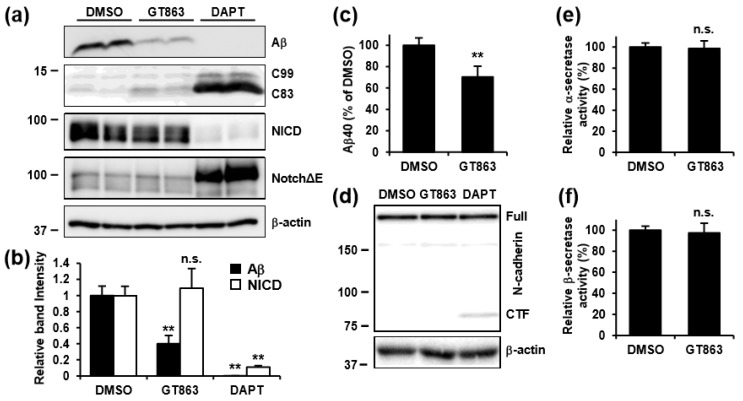
Effect of GT863 on activity of secretases. (**a**) CHO-APP/NotchΔE cells were treated with 3 μM GT863 or 1 μM DAPT for 48 h. Whole cell lysates were immunoblotted with appropriate antibodies as indicated. (**b**) Band intensities were quantified by densitometric scanning, relative intensity being shown. Mean ± SD *n* = 3, ** *p* < 0.01, n.s. = not significant. (**c**) SH-SY5Y cells were treated with 3 μM GT863 for 24 h. Secreted Aβ40 in conditioned media was measured by ELISA. *n* = 3. ** *p* < 0.01. (**d**) SH-SY5Y cells were treated with 3 μM GT863 or 1 μM DAPT for 24 h. Whole cell lysates were immunoblotted with appropriate antibodies specific for N-cadherin or β-actin. (**e,f**) α-secretase (**e**) or β-secretase (**f**) activity in SH-SY5Y cells treated with 3 μM GT863 for 24 h was measured and normalized to control cells. *n* = 3. ** *p* < 0.01.

**Figure 5 cells-09-00349-f005:**
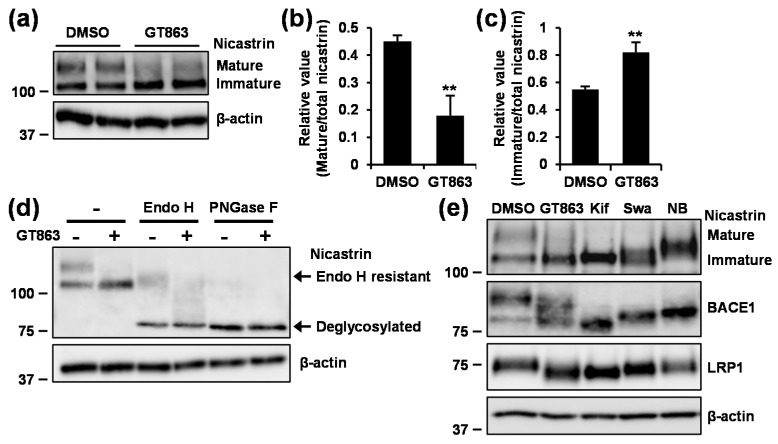
Effect of GT863 on nicastrin maturation. (**a–d**) CHO-APP/NotchΔE cells were treated with 3 μM GT863 for 48 h. (**a**) Whole cell lysates were immunoblotted with appropriate antibodies specific for nicastrin or β-actin. (**b,c**) Band intensities of mature (**b**) and immature (**c**) nicastrin were quantified by densitometric scanning, with the relative value being shown. Mean ± SD *n* = 3, ** *p* < 0.01, n.s. = not significant. (**d**) Whole cell lysates were untreated or treated with endoglycosidase H (Endo H) or peptide-N-glycosidase F (PNGase F), and immunoblotted with appropriate antibodies specific for nicastrin or β-actin. (**e**) CHO-APP/NotchΔE cells were treated with 3 μM GT863, 1 μg/mL kifunensine (Kif), 1 μg/mL swainsonine (Swa), or 1 mM *N*-butyl-deoxynojirimycin (NB-DNJ) (NB) for 48 h. Whole cell lysates were immunoblotted with appropriate antibodies as indicated. LRP1: low-density lipoprotein (LDL) receptor-related protein 1; BACE1: β-site APP cleaving enzyme 1.

**Figure 6 cells-09-00349-f006:**
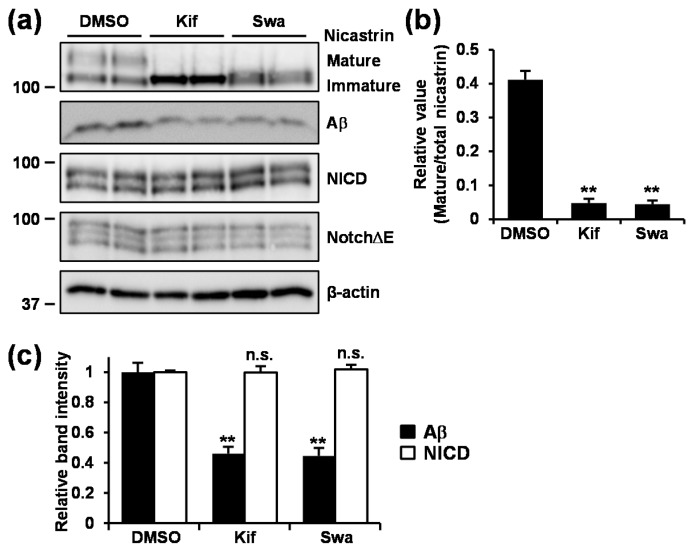
Effect of kifunensine or swainsonine treatment on Aβ and NICD production. (**a**–**d**) CHO-APP/NotchΔE cells were treated with 1 μg/mL kifunensine (Kif) or 1 μg/mL swainsonine (Swa) for 48 h. (**a**) Whole cell lysates were immunoblotted with appropriate antibodies as indicated. (**b**–**d**) Band intensities of nicastrin (**b**), and Aβ and NICD (**c**) were quantified by densitometric scanning, relative value being shown. Mean ± SD *n* = 3, ** *p* < 0.01, n.s. = not significant.

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
