# Peer review of "Curcumin Derivative GT863 Inhibits Amyloid-Beta Production via Inhibition of Protein N-Glycosylation"

_cells, 2020, doi:10.3390/cells9020349_

Round 1

Reviewer 1 Report

The manuscript "Curcumin Derivate GT863 Inhibits Amyloid-Beta production via Inhibition of protein N-Glycosylation" represents an excellent scientific work. This manuscript describes a study effects of Curcumin derived treatments  on Amyloid Beta production  l. This work is clearly designed and focused and presents interesting results about respective regulation at cellular and molecular level.

However, there is  a comment required to be corrected:

In figure 1 a you described cell viability after treatment with curcumin derivates.  The diagram is not very clear. It would be better to rework the diagram

Author Response

We appreciate the Reviewer’s valuable comments. As suggested by the Reviewer, we have revised figure 1a.

Reviewer 2 Report

I think its a great work and relevant to publish in the Cell Journal provided that the authors may respond to the comments below.

Please include similar and recent studies to boost your systems' originality in the introduction  Why Specifically the modification changed better results should be clearly mentioned.

Author Response

Please include similar and recent studies to boost your systems' originality in the introduction 

Our response:

We appreciate the Reviewer’s valuable comments. Based on the reviewer’s comment, we have revised the introduction section and we have cited new references as follows.

Line 65:

While curcumin derivatives have been the focus of studies seeking to develop inhibitors of Ab and tau aggregation, and have been the focus of studies seeking to develop imaging probes for detection of Ab and tau fibrils, there has been very little investigation into whether curcumin derivatives might serve as inhibitors of Ab production [13-19].

Why Specifically the modification changed better results should be clearly mentioned.

Our response:

Based on the reviewer’s comment, we have revised the Discussion section as follows.

Line 320:

We also speculate that the good inhibitory effect of GT863 on Ab production as compared with CU6/CNB-001, GT855, or GT857 may have been due to presence of the hydrogen in the pyrazole moiety of GT863 at the location where there is a benzene or nitrobenzene group in the pyrazole moiety of CU6/CNB-001, GT855, and GT857.

Reviewer 3 Report

The manuscript entitled “Curcumin Derivative GT863 Inhibits Amyloid-Beta Production via Inhibition of Protein N-Glycosylation” deals with the effectiveness of the use of the curcumin derivative GT863/PE859 in the reduction of Aβ secretion by supressing γ-secretase mediated cleavage and N-linked glycosylation, including that of the 26 γ -secretase subunit nicastrin studied in vitro on Chinese hamster ovary (CHO) cells that stably expressed wild-type human APP751.

The techniques used in the present manuscript, namely WST-8 assay to assess cell viability; sandwich ELISA for Aβ secretion; immunoblotting and γ -secretase analyses are adequate for its goal.

The manuscript is well written and easy to understand. The references used in the manuscript are recent and are adequate.  Regarding the novelty of the manuscript as far as I am concerned the authors have worked with the curcumin derivative GT863 in other publications in the last few years, but the present manuscript suggests a mechanism of action downregulating N-glycosylation.

In my opinion, the findings shown here are interesting for a broader community and deserve to be published.

Author Response

We appreciate the Reviewer’s valuable evaluation.